# Unveiling Let-7a’s Therapeutic Role in Ewing Sarcoma Through Molecular Docking and Deformation Energy Analysis

**DOI:** 10.3390/cimb47110948

**Published:** 2025-11-14

**Authors:** Mubashir Hassan, Amal Malik, Saba Shahzadi, Andrzej Kloczkowski

**Affiliations:** 1The Steve and Cindy Rasmussen Institute for Genomic Medicine, Nationwide Children’s Hospital, Columbus, OH 43205, USA; 2Institute of Molecular Biology and Biotechnology, The University of Lahore (Defense Road Campus), Lahore 54590, Pakistan; amalmalik.96@gmail.com; 3Department of Pediatrics, The Ohio State University, Columbus, OH 43205, USA

**Keywords:** Ewing sarcoma, Let-7a, miRNAs, EWS-FLI1 protein, EWSR1, docking

## Abstract

Ewing sarcoma is a pediatric malignant cancer that usually develops in bones and soft tissues. The current study investigates the function of hsa-let-7a as a target molecule in the pathophysiology of Ewing sarcoma using computational approaches. To anticipate complementary sites, miRNA and mRNA sequences were retrieved from the miRBase and NCBI databases. The three-dimensional structures of both hsa-let-7a and mRNA_EWSR1 were predicted through MC-Fold and RNAComposer, respectively. Furthermore, online HNADOCK and PatchDock docking servers were utilized to check the docking energy values and interactive behavior between miRNA and mRNA. The generated docked results showed good binding score values and interaction profiles between nucleotides of hsa-let-7a and mRNA of EWSR1. Moreover, both docking complexes were also studied using anisotropic network model analysis, which involved plotting correlation, inter-nucleotide distance fluctuations, and deformation energy graphs. The predicted heatmap graph also highlighted the significance of hsa-let-7a in various cellular signaling pathways, which may be interconnected with Ewing sarcoma, making it a potential therapeutic target. Together, this study offers computational insights that highlight hsa-let-7a as a promising therapeutic candidate for Ewing sarcoma, based on miRNA-driven predictive modeling.

## 1. Introduction

Ewing sarcoma (ES) is a rare type of cancer that affects bones and the soft tissues surrounding them [1]. ES is the second-most common malignant bone tumor occurring in children and young adults [2]. The metastatic behavior of ES showed that it may affect the entire skeletal system; however, 45% of cases are located in the lower extremities, 20% in the pelvis, 13% in the upper extremities, the axial skeleton, and ribs, and 2% in the facial region, respectively [3]. Currently, the exact pathogenesis of ES remains unclear. It has been observed that ES is primarily (85%) caused by the expression of EWS/FLI through unique chromosomal translocations in primary cells, leading to growth arrest or cell death [4]. The differentiation defects in this disease result from the increased expression of EWS/FLI in primitive cells [5].

Multiple cellular therapeutic targets have been used to properly understand the etiology of ES [6]. Different proteins, such as RNA Polymerase II, hsRBP7, CREB binding protein, Matrix Metalloproteinase (MMP-1), Nuclear Receptor Subfamily 0 Group B Member 1 (NR0B1), and NK2 Homeobox 2 (NKX2), are directly or indirectly involved in the ES through transcriptional regulatory mechanisms. Besides all the protein machinery, microRNAs (miRNAs) are also active players in controlling the prevalence of ES through transcriptional regulation [5]. miRNAs are endogenous, small non-coding RNAs that function in the regulation of gene expression [7,8]. Compelling evidence has demonstrated that miRNA expression is dysregulated in human cancer through various mechanisms [9,10], including amplification or deletion of miRNA genes, abnormal transcriptional control of miRNAs, dysregulated epigenetic changes, and defects in the miRNA biogenesis machinery [11]. Moreover, miRNAs can be utilized as a therapeutic treatment option for various diseases that have been emerging in recent times, with a vast range of applications, including use in vaccines [12]. Prior research has shown strong indications that let-7 family members (let-7a, let-7b, let-7g, let-7e, and let-7f) are significantly involved in tumor suppression activities in various cancers [13,14]. Moreover, miRNAs are shown to be involved in gene regulation, cell adhesion, and muscle formation [15]. The let-7 family members play a significant role in the prevalence of ES through the transcriptional and translational activity of various genes, such as High-mobility group AT-hook 2 (HMGA2) [8,16,17].

miRNAs, specifically let-7a-3p, have a good correlation with the EWSR1 gene (a well-known target for ES); therefore, a treatment can be developed and planned to slow down or control the prognosis of ES. Herein, a computational study has been designed to find the region complementarity sequences of the mRNA of EWSR1 against let-7a-3p through sequence alignment approaches. After that, a molecular modeling approach was employed to build the three-dimensional (3D) models of both mRNA and miRNAs using computational methods. Model confirmation and optimization were performed, and a molecular docking study was conducted to predict the orientation and interaction between the miRNA (let-7a-3p) and the mRNA (EWSR1), along with its predicted docking scores. Furthermore, mirPath was implemented to predict the KEGG pathway analysis of miRNAs.

## 2. Computational Methodology

### 2.1. Retrieval of miRNA Sequences from miRBase and miRDB

The precursor and mature sequences of miRNAs (let-7a-3p) were retrieved from miRbase (http://www.mirbase.org/: accessed on 29 October 2025) and miRDB (http://mirdb.org/: accessed on 29 October 2025). miRBase and miRDB are standard databases for mature and precursor miRNA sequences, along with their annotations. Both databases were used to verify the let-7a-3p sequences. In contrast, miRDB also provides a target score (69) of let-7a-3p with the EWSR1 gene.

### 2.2. National Center for Biotechnology Information (NCBI)

The mRNA sequence of EWSR1 was retrieved from the NCBI database (https://www.ncbi.nlm.nih.gov: accessed on 29 October 2025), having accession number (NM_001163287), and the sequence was retrieved in the FASTA format.

### 2.3. miRNA and mRNA Model Predictions

The 2D structures of let-7a and mRNA_EWSR1 were generated by the MXfold2 server (http://www.dna.bio.keio.ac.jp/mxfold2/predict: accessed on 29 October 2025). Furthermore, MC-Fold (https://major.iric.ca/MC-Fold/: accessed on 29 October 2025) and MC-Sym (https://major.iric.ca/MC-Sym/: accessed on 29 October 2025) online servers were employed to produce their 3D models. The designed miRNA models were downloaded in PDB format and used for further structural analysis. For mRNA model prediction, RNAComposer (http://rnacomposer.cs.put.poznan.pl/: accessed on 29 October 2025), an online platform, was used to predict the three-dimensional (3D) structures of mRNA molecules of EWSR1. The mRNA sequence, retrieved from the NCBI server, was incorporated into the RNAComposer, and a predicted 3D structure of the mRNA was generated using default parameters. Moreover, the UCSF chimera tool (1.10.1) was used to visualize the miRNA and mRNA models, and their graphical images were depicted [18]. In this study, our analysis is based on RNAComposer (1.0), a well-established and extensively validated platform within the RNA research community. Recently, several new methods for predicting the 3D structures of RNAs and their interactions with proteins and nucleic acids, based on deep learning, have been proposed, including AlphaFold 3 [19], RhoFold+ [20], trRosettaRNA [21], and RoseTTAFoldNA [22]. In a recent study [22], a comparison was performed between RNAComposer, Rosetta FARFAR2, and AlphaFold 3, and it was found that RNAComposer’s predictions for the 3D structure of a malachite green aptamer are closer to its crystal structure. Since the number of crystallographically solved RNA structures is still relatively small, deep-learning-based methods have their limitations. That is clearly evident from the most recent comparison of the accuracy of RNA structure predictions in CASP16 experiments. The comparison data shown on the CASP16 website https://predictioncenter.org/casp16/zscores_rna.cgi: accessed on 29 October 2025 shows that the predictions of 3D RNA structures by the AlphaFold 3 server were rather poor. The AlphaFold server was classified at 25th position among 64 participants measured by Rank AVG Zscore. We use this Rank average Zscore for comparison, instead of Rank SUM Zscore, since AlphaFold 3 submitted predictions for all RNA targets, while many groups skipped some of the targets, which affects their summary Z scores. AlphaFold3, though capable of predicting joint structures and interactions with RNA using a diffusion-based model for atom coordinates, sometimes violates chirality, predicts only static structures, failing to capture dynamic behavior, and can introduce false predictions in disordered regions, often with low confidence [19,23]. RoseTTAFoldNA, an extension specifically designed for nucleic acid complexes, offers higher accuracy in confident predictions for various protein–nucleic acid complexes but may exhibit lower overall accuracy compared to AlphaFold2 for proteins, and has memory limitations that initially restrict sequence input [22]. Finally, RhoFold+, designed for rapid 3D prediction of single-chain RNAs by leveraging RNA language models, struggles with large and complex RNA structures, especially those with multiple helices or pseudoknots, and can suffer from training data biases and memory constraints, limiting its application to longer sequences [24,25].

We selected the RNAComposer tool for its robust performance, reproducibility, and relevance to our experimental objectives. While we acknowledge that structural predictions may vary between algorithms, our primary focus was to explore the biological implications of the predicted mRNA structure as derived from a consistent methodological framework. A comparison of our predicted 3D RNA model with predictions from AlphaFold 3, RhoFold+, and trRosettaRNA methods is presented in the Results section.

### 2.4. Molecular Docking of Mature miRNA and mRNA

Molecular docking is the most effective computational approach for evaluating the interaction between biological molecules [26,27]. The HNADOCK docking server (http://huanglab.phys.hust.edu.cn/hnadock/: accessed on 29 October 2025) was employed to verify its binding pattern with predicted genomic motifs and observe their conformation behavior. The miRNA–mRNA docking is based on a hybrid algorithm that combines template-based modeling and an ab initio free docking method. In the general procedure, 3D structures of both mRNA of EWSR1 and miRNA (let-7a-3p) were uploaded in PDB format and run docking with maximum search space. The top ten docking complexes were downloaded and analyzed in Discovery Studio and Chimera, respectively.

To confirm the accuracy of our miRNA–mRNA binding interaction pattern and the accuracy of our docking results, the PatchDock docking algorithm was used, based on shape complementarity principles (https://bioinfo3d.cs.tau.ac.il/PatchDock/: accessed on 29 October 2025). This method simultaneously addresses the issues of flexibility and scoring of solutions generated by fast rigid-body docking algorithms. We obtained 100 potential docking candidates; PatchDock refines and scores them according to an energy function, spending about 3.5 s per candidate solution. The bonding interaction pattern of all docking servers between miRNA and mRNA (EWSR1) was keenly observed by using Discovery Studio (4.1) [28] and UCSF Chimera 1.10.1 [18], respectively. Finally, the Anisotropic Network server was used to check the correlation and energy deformation behavior of both docked complexes [29].

HNADOCK has clear advantages over PatchDock for miRNA–mRNA docking, as it is designed explicitly for RNA–RNA interactions, incorporates RNA sequence and secondary structure inputs, and utilizes the RNA-optimized DITScoreRR scoring function to enhance prediction accuracy. It is particularly well-suited for miRNA–mRNA modeling due to its high performance (about 71.7% success rate in top predictions) and ability to accommodate binding site constraints. The broad shape complementarity technique used by PatchDock, in contrast, may produce more false positives and necessitate manual post-processing, despite being capable of analyzing RNA structures. HNADOCK is a better option for studying miRNA-mediated gene regulation because it provides more contextually aware and biologically relevant data for nucleic acid interactions, even though PatchDock is helpful for broad structural docking tasks.

Furthermore, the LZerD protein docking server (https://lzerd.kiharalab.org/about/: accessed on 29 October 2025) was used to examine the interaction profile between EWS-FLI1 and wild-type EWSR1 proteins. The 3D structure of EWSR1 was accessed from the PDB, with ID 2CPE, whereas the wild-type EWSR1-FLI1 structure was accessed from the AlphaFold dataset (AF-F1JVV7-F1) and utilized in the PDB, with ID 2CPE. The wild EWSR1-FLI1 structure was also accessed from the AlphaFold dataset (AF-F1JVV7-F1) and used in the docking experiment.

### 2.5. KEGG Pathway Prediction of miRNAs

mirPath is the new version of the DIANA-miRPath web server (http://www.microrna.gr/miRPathv2: accessed on 29 October 2025). DIANA-miRPath performs miRNA pathway analysis, providing accurate statistics while accommodating advanced pipelines. mirPath was implemented to predict the Kyoto Encyclopedia of Genes and Genomes (KEGG) pathway and gene ontology (GO) prediction of miRNAs [30]. miRTargetLink 2.0, an online server, was used to investigate the interactive behavior of has-let-7a against different target genes and their corresponding pathway networks [31].

## 3. Results and Discussion

### 3.1. Retrieval of mRNA of EWSR1

The mRNA sequence of EWSR1 has been selected based on sequence complementarity alignment (Figure 1A), and the predicted mRNA model of EWSR1 is depicted in Figure 1B.

For comparison, we computed the mRNA sequence of EWSR1 predicted by three different AI-based tools: AlphaFold 3, RhoFold+, and trRosettaRNA. We observe that each method predicts a different structure, with the AlphaFold 3 prediction significantly different from all other structures, and the RhoFold+ and trRosettaRNA structures bearing more similarity to our structural model, as shown in Figure 2.

### 3.2. Sequence and Structural Analysis of Generated Models

The size of the family of miRNA depends upon the species being observed and studied. There have been three separate precursor sequences found to produce the hsa-let-7a, hsa-let-7a-1, hsa-let-7a-2, and hsa-let-7a-3, respectively. Precursors from two different genomic locations produce the hsa-let-7f sequence, which are hsa-let-7f-1 and hsa-let-7f-2. Those involved in ES include hsa-let-7a, hsa-let-7b, hsa-let-7e, hsa-let-7f, and hsa-let-7g. Their involvement is evident through the deregulation of these miRNAs in more than 90% of patients with ES, as observed through target gene prediction conducted for the deregulated miRNAs in over 90% of patients with ES [16]. It has been studied that let-7a is one such miRNA that is a direct target of EWS-FLI1 activity. The let-7a precursor sequence contains two mature miRNA sequences, 3p (containing 21 nucleotides) and 5p (containing 22 nucleotides). We are more concerned with the 3p sequence, as it aligns with greater complementarity to the mRNA of the EWSR1 protein than the 5p sequence, and provides us with more docked models (Table 1). Whereas the alignment of all five miRNAs and the predicted model of let-7a have been depicted in Figure 3 and Figure 4.

Through the research study conducted by Claudio De Vito and Nicolo Riggi [4], it has been noted that the tumorigenic capacity of ES tumor cells is connected to the direct suppression of hsa-let-7a by EWS-FLI-1. It was also then determined that the mechanism by which hsa-let-7a expression controls the growth of ES tumors is mediated by the HMGA2 protein-coding gene, which is a direct target of let-7a. As hsa-let-7a is overexpressed and HMGA2 is repressed in turn, both may work to block the tumorigenicity of ES tumor cells. Hence, there can be therapeutic use of overexpression and use of let-7a miRNA against the aggressive malignant tumor, forming mRNA and proteins, such as those in ES.

### 3.3. Molecular Docking Analysis

#### 3.3.1. HNADOCK Docking

The results of this molecular docking output are the docked structures of the let-7a-3p sequence and the messenger RNA of EWSR1, providing the ligand RMSD (Root Mean Square Deviation) values in angstroms. This value dictates the correctness of the molecular docking geometry by measuring the deviation of the ligand from the reference position in the suggested complex docked structure. The docking study results predict the top ten docking complexes, which exhibited good docking scoring values such as −277.63, −259.61, −259.30, −257.72, −251.00, −250.45, −247.41, −247.13, −244.98, and −238.91, respectively. The comparative results showed that docking complex 1 exhibited better scoring values than the other docking complexes. The smaller or more negative the docking score, the better the affinity or stronger the binding (Figure 5). The top docking complex of let-7a with the mRNA of EWSR1 displayed interactive behavior within a specific region through complementary nucleotide sequence interactions (Figure 6).

#### 3.3.2. PatchDock Docking

PatchDock is another docking tool available online and easy to use for creating 3D models of two molecules, such as proteins, DNA, peptides, and drugs. In our docking results, the docking complexes exhibited good scoring values: 15,716, 15,300, 15,108, 15,020, 14,992, 14,876, 14,728, 14,714, 14,714, and 14,660, respectively. The first docking complex shows the best scoring value as compared to other docking complexes. Higher docking scoring values justify the good conformational behavior against the target proteins (Figure 7).

#### 3.3.3. Interaction Behavior

Figure 8 displays the interactive behavior of mature let-7a-3p and the mRNA of EWSR1 in the docking complex. The generated results showed that let-7a-3p binds to the mRNA of EWSR1 at a specific region through a complementary nucleotide sequence. A total of 21 nucleotides (cuauacaaucuacugucuuuc) in let-7a-3p take part in the interaction with the mRNA of EWSR1. That suggests that these nucleotides form a bond with the complementary sequence from 559-G to 580-A of the mRNA of EWSR1.

However, docking tools often provide a simplified view of miRNA–mRNA interactions, overlooking critical biological features such as RNA secondary structure, protein cofactors, and cellular localization. While useful for initial screening, docking can be complemented with experimental or transcriptomic data for meaningful interpretation. In addition to our docking results, literature-based evidence has been added to support the anticipated interaction between hsa-let-7a and EWSR1 mRNA. Interestingly, luciferase assays, qRT-PCR, microarray, and Western blot analysis have experimentally validated the targeting of hsa-let-7a-5p to EWSR1, which also supports our docking results [4].

#### 3.3.4. EWS-FLI1 Docking Interaction

EWS-FLI1 is considered a causative oncogene in ES, meaning it plays a crucial role in the development and progression of this cancer. It has been shown to transform cells, promote proliferation, and is necessary for the tumor’s ability to form tumors [32,33]. The retrieved EWS-FLI1 structure exhibited good docking scoring values and an interaction profile similar to that of the wild-type EWSR1. The generated docked complexes were ranked based on their scoring values (GOAP: −36,355.52, DFIRE: −24,890.96, and ITScore: −11,500.98). Among all the best docking complexes, the one depicted in Figure 9 is the best. Two hydrogen bonds have been observed at different residue positions between the molecules of both proteins. In EWS-FLI1, Asp431 and Gln383 form hydrogen bonds with Arg309 and Gln422 (EWSR1), with bond lengths of 3.44 Å and 3.04 Å, respectively.

miRNAs regulate gene expression post-transcriptionally, generally binding to the 3′-UTR of their target mRNAs and repressing protein production by destabilizing the mRNA and inducing translational silencing [34]. In comparative docking results, it has been observed that adenine (A) at position 8 in let-7a formed hydrogen bonds with cysteine (C) at positions 59 and 60, with bond lengths of 3.02 and 2.01 Å, respectively. Furthermore, uracil (U) at position 11 in Let-7a forms hydrogen bonds with adenine (A) and guanine (G) at positions 62 and 63, having bond distances 2.33 and 2.78 Å, respectively (Figure 10).

### 3.4. Anisotropic Network Model Analysis

#### Correlation Analysis by Two-Dimensional Matrices

The anisotropic network model results show fluctuations of nucleotides in mRNA with correlations represented in different colors. The generated models of mRNAs of both docking complexes are represented in red, blue, and white colors, respectively. The correlated nucleotides are highlighted in red, whereas anti-correlated nucleotides are highlighted in blue. Additionally, weak correlations are shown in light colors. The comparative analysis revealed that the HNADOCK docking complex exhibits a good correlation pattern among nucleotides, compared to the PatchDock complex, which ensures a favorable binding conformational behavior of let-7a with the mRNA (Figure 11A,B). The distribution of the 20 dominant eigenvalues in bar graphs represents the visual inspection for the relative contribution of each mode and possible degeneracy between the modes in both docking complexes (Figure 12A,B).

### 3.5. Inter-Nucleotide Distance Fluctuations and Deformation Energy

The inter-nucleotide distance is a novel numerical profile used to explore the correlation structure of DNA [28], and deformation energy is proportional to the sum of the square fluctuations of all interacting nucleotides [29]. The generated results showed that nucleotides of mRNA depicted different deformation energy (kcal/mol) peaks in the deformation graph. In the HNADOCK docking complex, the nucleotides of mRNA at positions 1–16 showed a slight increasing trend with 0.018 (kcal/mol), whereas nucleotides at positions 30–60 showed a decreasing trend in the graph. Nucleotides at positions 30–60 exhibited a simultaneous decreasing and increasing trend in the graph line, with a slight increase in energy value of 0.036 kcal/mol. Moreover, the nucleotides around positions 62–67 showed the highest fluctuation peak and energy values >0.072 < 0.09 kcal/mol Moreover, the nucleotides at positions 62–67 showed the highest fluctuation peak and energy values >0.072 < 0.09 kcal/mol. Furthermore, two additional increasing peaks were observed in the deformation energy graph, located around nucleotides 76 and 89, with values of 0.036 and 0.054 kcal/mol, respectively. After that, all the nucleotides in the mRNA structure remained stable, with slight fluctuations, exhibiting peaks with an energy of 0.018 kcal/mol (Figure 13A). In the PatchDock complex, the nucleotides of mRNA at positions 1 to 18 showed an increasing trend in graph peaks, with a range of 0 to 0.024 kcal/mol. A similar up-and-down fluctuation was observed at nucleotide sites 20–45, with a similar energy value. The three major peaks were reported between the clump of 50–90 nucleotides, with the highest energy value of 0.04 kcal/mol. Moreover, for nucleotides at positions 90–140, a steady behavior of the graph was observed, with an energy range of 0.08–0.018 kcal/mol (Figure 13B). The comparative results showed that the binding of miRNA to mRNA causes a slight conformational change, silencing the EWSR1 gene. As a result, the formation of the EWS protein is disrupted, which may impair the formation of the EWS-FLI-1 complex, a key factor in the development of ES.

### 3.6. miRNA KEGG Pathways Analysis

A heatmap is a data visualization technique that shows the magnitude of a cellular phenomenon and pathways associated with respective genes. The computed results show the significance of the let-7 family of miRNAs (hsa-let-7a, hsa-let-7b, hsa-let-7e, hsa-let-7f-1, and hsa-let-7g) in various cellular pathways (Figure 14A). These results indicate that hsa-let-7a-3p/5p, hsa-let-7b, and hsa-let-7f-1 are directly involved in the ECM-receptor interaction pathway (an upregulated gene-enriched signaling pathway), which is involved in the processes of tumor shedding, adhesion, degradation, movement, and hyperplasia [35]. Among all miRNAs, hsa-let-7a and hsa-let-7b showed a higher propensity for interaction with the ECM-receptor pathway, which provides a clear indication of the involvement of both miRNAs in sarcoma pathways. Moreover, hsa-let-7a also showed a little association with the cell cycle and Hippo signaling pathways. The hsa-let-7e and hsa-let-7g did not show any association with any cellular signaling pathways.

In targeted pathway enrichment analysis, it is demonstrated that hsa-let-7a-3p/5p, hsa-let-7b-3p, and hsa-let-7f-5p are involved in more targeted pathways than hsa-let-7e and hsa-let-7g. Hsa-7a-3p/5p, hsa-7b-3p, and hsa-7f-5p showed high prevalence in relation to Hippo signaling pathways, small cell lung carcinoma, colorectal cancer, and endocytosis, respectively. Moreover, hsa-7b-3p and hsa-7f-5p also have associations with the TG beta, P53, MAPK, and Foxo signaling pathways, as well as glioma, bladder cancer, chronic myeloid leukemia, hepatitis B, and the cell cycle (Figure 14B). It has been observed that these signaling pathways have a direct influence on the insurgence of ES by activating and deactivating different cellular molecules [36,37,38]. The comparative results of miRNAs showed that hsa-let-7a and hsa-let-7f have a stronger association with various signaling pathways and could play a significant role in the pathogenesis of ES.

Table 2 highlights key findings from the KEGG pathway enrichment analysis of let-7 family miRNAs targeting EWSR1 mRNA. For each miRNA–pathway pair, it lists how many target genes are involved (gene count), the statistical significance after correction for multiple testing (adjusted *p*-value), and a calculated enrichment score that reflects the strength of the association. These values offer a clearer understanding of which pathways are most relevant and help support the visual trends shown in Figure 14.

In ES, its modulation of TGF-β signaling could inhibit tumor-promoting cues often exploited by EWS-FLI1. Similarly, interference with MAPK signaling may reduce proliferative and survival signals, while reactivation of p53-related pathways could restore mechanisms of apoptosis and cell cycle regulation [39,40]. Previous studies have demonstrated that the let-7 family of miRNAs can regulate components of these signaling cascades, particularly through direct targeting of RAS genes and modulation of the PI3K/Akt pathway, thereby influencing cell proliferation, differentiation, and tumorigenesis [41,42]. Additionally, EWSR1 has been implicated in transcriptional regulation and chromatin remodeling, processes that intersect with these pathways in various cancers [43].

### 3.7. hsa-let-7a-3p Target Gene and Target Pathway Networks

To address one-to-many or many-to-many interactions, miRTargetLink 2.0 aims to facilitate the systematic study of miRNAs in relation to their target genes and pathways [31]. The generated results showed that hsa-let-7a-3p and hsa-let-7a-5p have interactions with different genes and may control their downstream signaling pathways. The list of predicted genes, along with experimental citations, has been tabulated in Table 3.

### 3.8. Mechanistic Pathway of ES

Hsa-let-7a, along with its family members, plays a significant role in cell differentiation and tumor suppression by silencing numerous genes that encode oncogenic proteins [74]. Let-7a is a non-coding transcript that undergoes multiple processing steps, initiated by RNA polymerase-II-mediated transcription to generate a primary pri-let-7a [75]. The pri-let-7a is processed by the multiprotein microprocessor complex, which includes Drosha and DGCR8/Pasha, to produce a ∼70-nucleotide (nt) precursor called pre-let-7a. Furthermore, pre-let-7a is subsequently exported by Exportin-5 from the nucleus to the cytoplasm, where the Dicer complex further processes it to generate the mature let-7a (22 nt) [75]. This mature sequence, in turn, may form a complex with the mRNA of EWSR1 to silence this gene and disrupt the formation of the EWS protein, thus also impairing the formation of the EWS-FLI-1 complex that occurs in ES [4]. The overall mechanistic pathway is illustrated in Figure 15.

## 4. Conclusions

The EWS-FLI1 is a well-known responsible gene involved in the governance of ES through a direct complementary target site for hsa-let-7a binding. Therefore, in this current research, 3D models of both hsa-let-7a and the mRNA of EWSR1 have been built and analyzed to check their conformational binding sites. The miRNA–mRNA docking results indicated that hsa-let-7a binds to the complementary sites of the mRNA in the EWSR1 gene, potentially silencing the oncogenic activity of EWS-FLI1. Furthermore, the binding behavior was confirmed through energy deformation analysis, which ensured that most nucleotides remained stable in the docking complexes, except for those around positions 62–67, which showed the highest fluctuation peak and deformation energy values of >0.072 to <0.09 kcal/mol. The KEGG pathway results showed that hsa-let-7a has an association with the TGF, MAPK, and P53 pathways, which are linked to ES pathogenesis. Based on computational results, it has been concluded that hsa-let-7a could be involved in the basic mechanism for augmenting EWS/FlI-1 in ES. Moreover, hsa-let-7a could also help to develop new biomarkers and miRNA-based drugs that can cure ES.

## Figures and Tables

**Figure 1 cimb-47-00948-f001:**
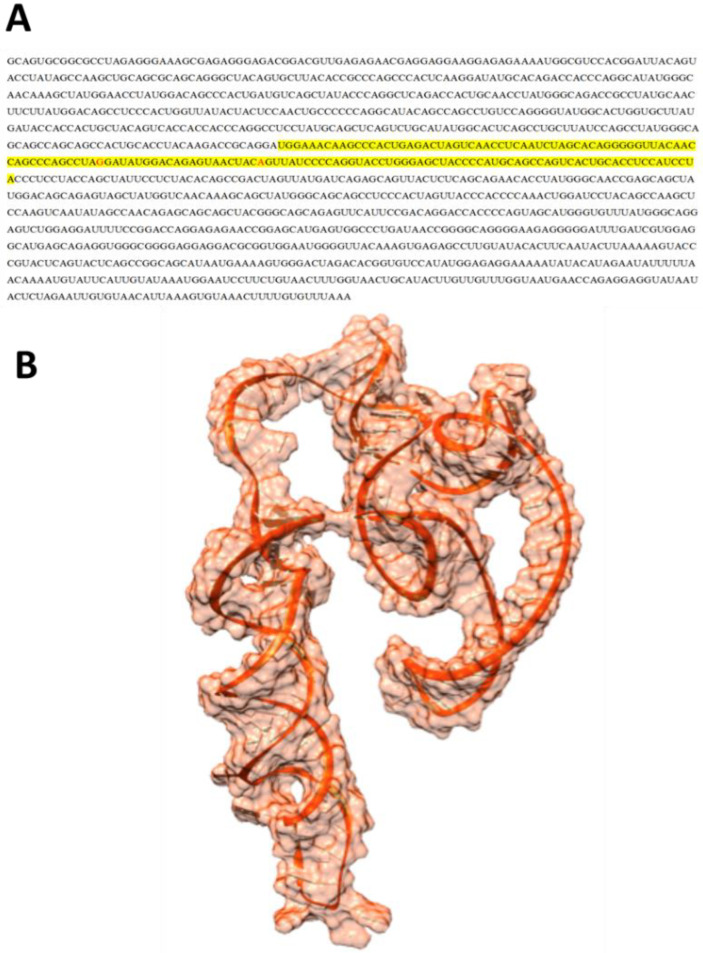
(**A**,**B**). mRNA sequence of EWSR1 with highlighted interaction portion and predicted model.

**Figure 2 cimb-47-00948-f002:**
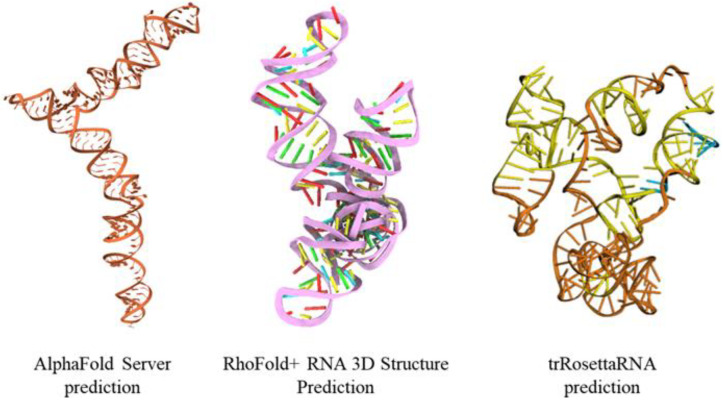
The three-dimensional structural model of mRNA sequence of EWSR1 predicted by three different AI-based tools: AlphaFold 3, RhoFold+, and trRosettaRNA.

**Figure 3 cimb-47-00948-f003:**
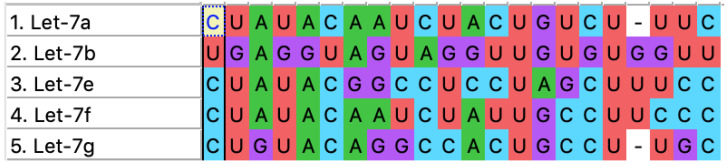
Alignment of the let-7 family of miRNAs. The fully aligned uracil among all miRNAs is shown in red at various positions.

**Figure 4 cimb-47-00948-f004:**
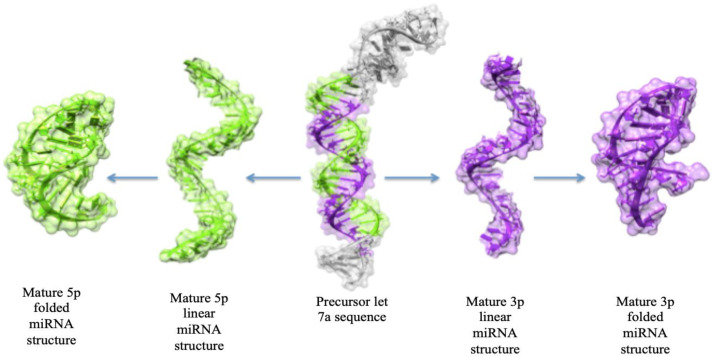
The predicted 3D conformations of precursor and mature let-7a miRNA structures. The right and left-sided arrows represent mature 3p and 5p miRs structures, respectively.

**Figure 5 cimb-47-00948-f005:**
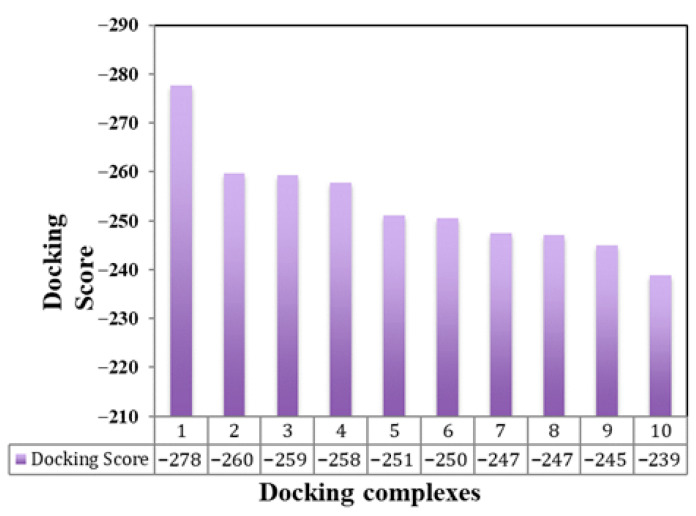
Docking scoring values between let-7a-3p- and mRNA-predicted models. This bar graph displays docking energy values for various predicted complexes formed between let-7a-3p and EWSR1 mRNA. The *x*-axis represents individual docked complexes, while the *y*-axis shows their corresponding docking scores. Lower scores indicate stronger and more stable interactions.

**Figure 6 cimb-47-00948-f006:**
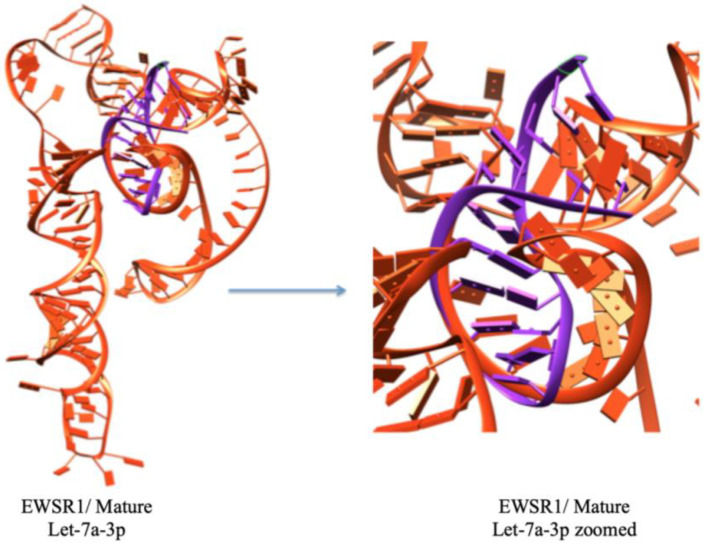
The best docking complex, where mRNA (EWSR1) is represented by orange color, and miRNA is depicted in purple color.

**Figure 7 cimb-47-00948-f007:**
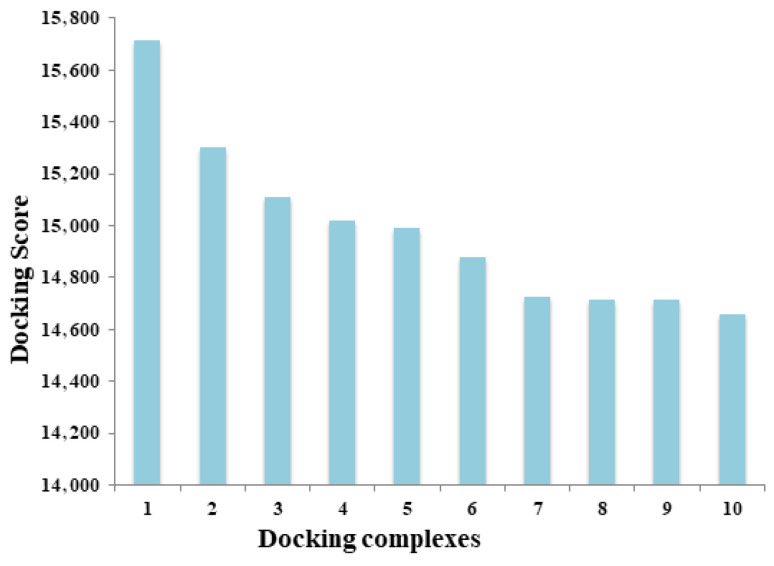
Docking scoring values between let-7a-3p and mRNA EWSR1. This bar chart represents a unique docked complex, and the height reflects its docking scores; higher values mean a tighter, more stable docking complex.

**Figure 8 cimb-47-00948-f008:**
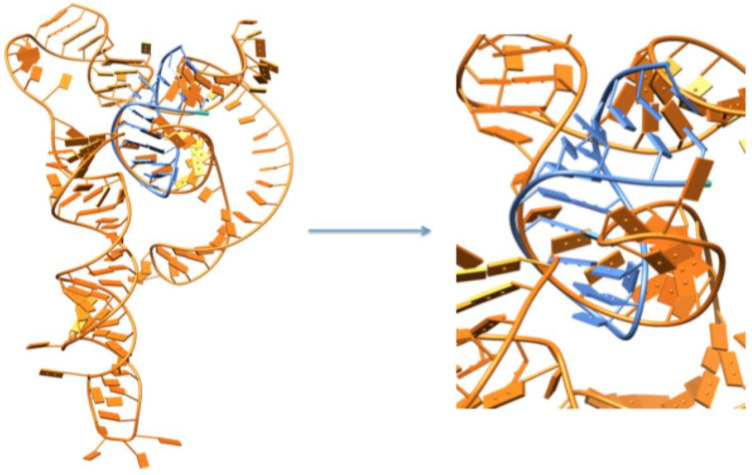
The best docking complex, where a light orange color represents the mRNA of EWSR1, and let-7a-3p is depicted in light blue.

**Figure 9 cimb-47-00948-f009:**
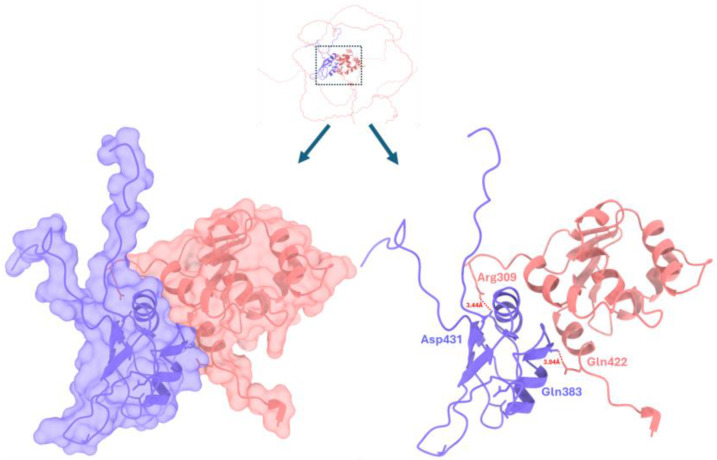
EWS-FLI/EWSR1 docking complex. The EWSR1 is represented in purple, whereas EWSR1-FLI1 is depicted in light brown, respectively. A dotted red line represents the hydrogen bond, and the distance is measured in angstroms (Å).

**Figure 10 cimb-47-00948-f010:**
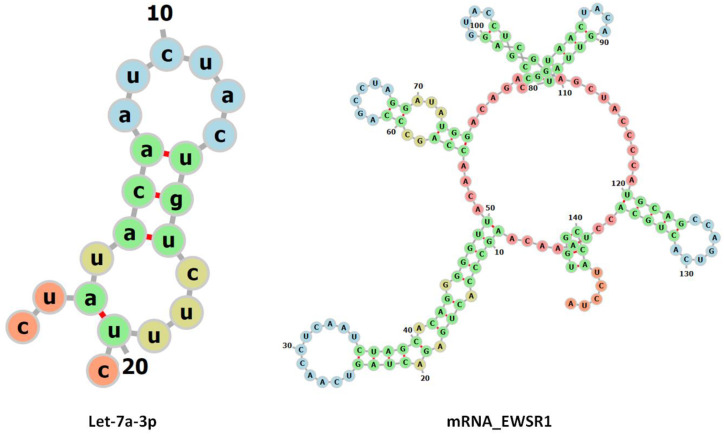
The 2D predicted structures of let-7a and the mRNA of EWSR1 are displayed.

**Figure 11 cimb-47-00948-f011:**
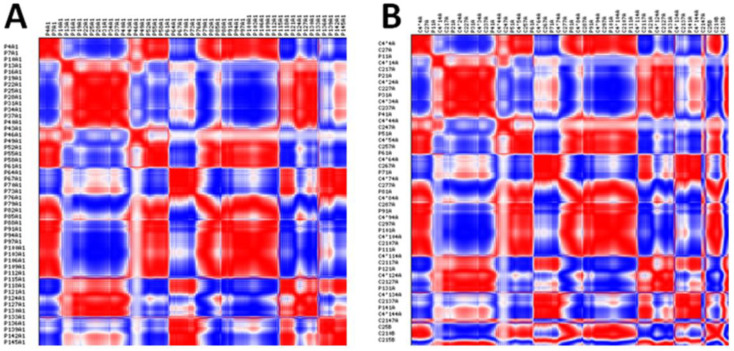
(**A**,**B**). Correlation of the mRNAs of both HNADOCK and Patchdock docking complexes. The correlated nucleotides are highlighted in red, whereas anti-correlated nucleotides are highlighted in blue.

**Figure 12 cimb-47-00948-f012:**
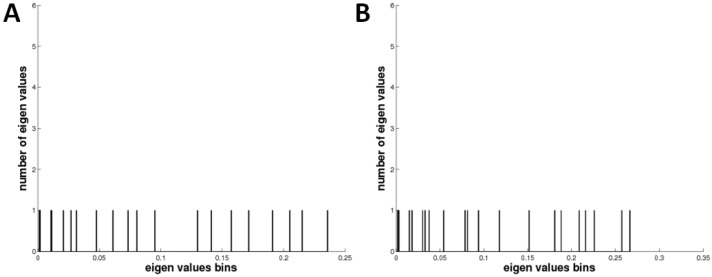
(**A**,**B**). Eigenvalue’s representation of both HNA and Patchdock docking complexes.

**Figure 13 cimb-47-00948-f013:**
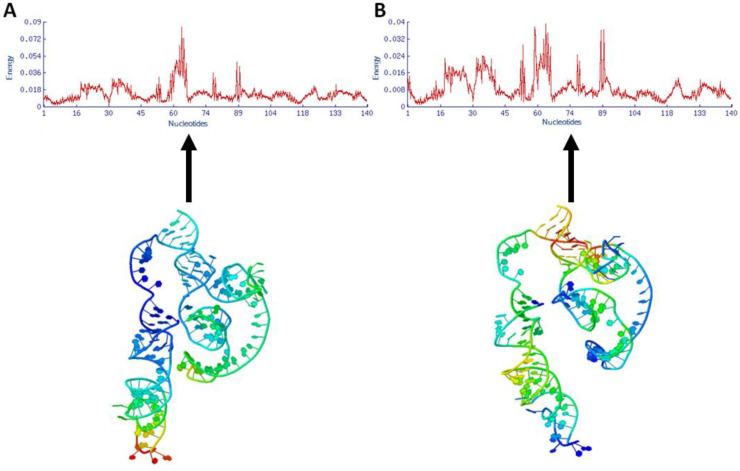
(**A**,**B**). The energy graphs of both docking complexes (HNADOCK (**A**) and PatchDock (**B**) of let-7a with the mRNAs of EWSR1). This bar graph illustrates the variation in docking energy across different nucleotide positions in the EWSR1 mRNA. Each bar reflects the energy score at a specific nucleotide, with the *y*-axis showing how strong or weak the interaction is. Peaks and dips in the graph highlight areas where the docking complexes are either more stable or less favorable, giving a clearer picture of how the mRNA behaves during binding.

**Figure 14 cimb-47-00948-f014:**
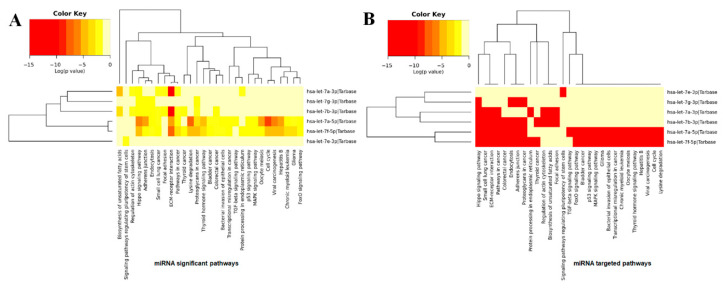
(**A**,**B**). miRNAs KEGG pathways analysis. The colors on the heatmap indicate the log (*p*-value) of the specific miRNAs and their involvement in the corresponding pathways.

**Figure 15 cimb-47-00948-f015:**
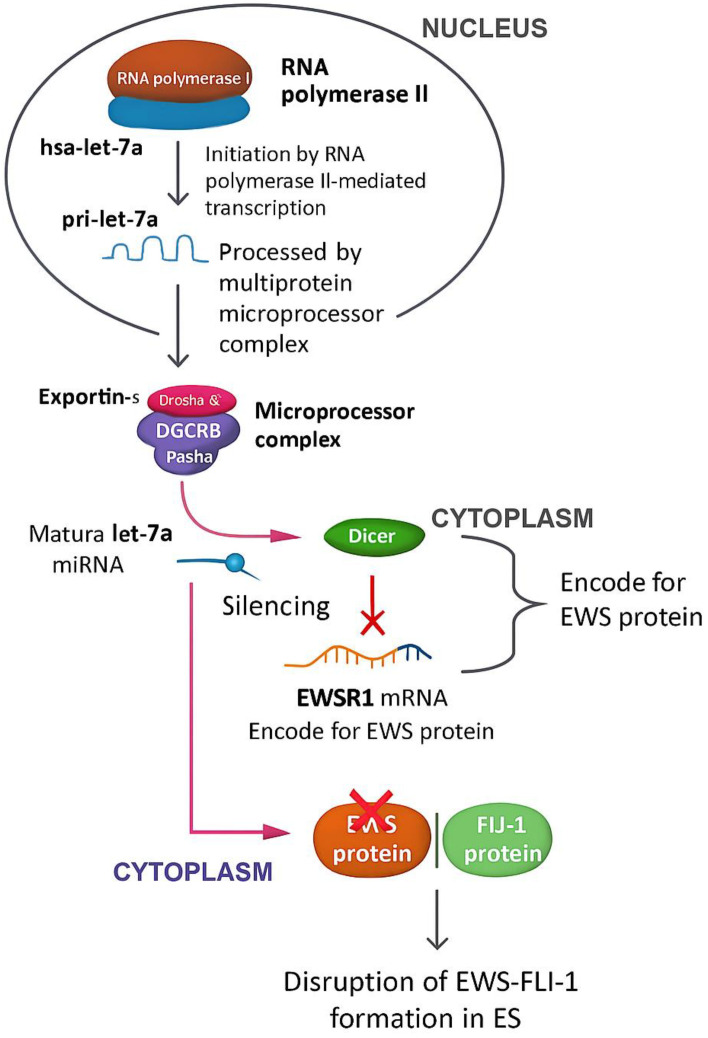
Mechanistic pathway of hsa-let-7a in ES. This diagram walks through the journey of hsa-let-7a from its origin to its role in suppressing tumor activity in Ewing sarcoma. It starts in the nucleus, where RNA polymerase II transcribes the primary let-7a transcript (pri-let-7a). That transcript is then trimmed down by the Drosha-DGCR8 microprocessor complex into a shorter precursor (pre-let-7a), which is exported to the cytoplasm by Exportin-5. Once in the cytoplasm, the Dicer complex processes it into the mature, functional let-7a microRNA. The mature let-7a then gets to work—binding to EWSR1 mRNA and silencing it. This silencing prevents the production of EWS protein, which is a critical component of the oncogenic EWS–FLI1 fusion complex found in Ewing Sarcoma. By disrupting this fusion, let-7a interferes with key cancer-driving mechanisms, highlighting its powerful role as a tumor suppressor and regulator of cell differentiation. The figure captures each step of this pathway, connecting molecular biology to therapeutic potential.

**Table 1 cimb-47-00948-t001:** Precursor and mature sequences of hsa-let-7 family.

miRNAs	Precursor miRNAs	Mature miRNAs
hsa-let-7a-1	ugggaugagguaguagguuguauaguuuuagggucacacccaccac auccu-uucugucaucuaacauaucaa-----------uagagggu	ugagguaguagguuguauaguu (5p) cuauacaaucuacugucuuuc (3p)
hsa-let-7b	cggggugagguaguagguugugugguuucagggcagugauguugccccucgg guccc-uuccgucauccaacauaucaa--------------------uagaa	ugagguaguagguugugugguu (5p)
hsa-let-7e	cccgggcugagguaggagguuguauaguuga----ggaggac ggaccccuuucgauccuccggcauauca-cuagaggaaccca	cuauacggccuccuagcuuucc (3p)
hsa-let-7f-1	ucagagugagguaguagauuguauaguugu---------gggguagugaaguc ccuuccguuaucuaacauaucaauagaggacuugucccauuuu	cuauacaaucuauugccuuccc (3p)
hsa-let-7g	aggcugagguaguaguuuguacaguuugagggucu-augauaccac accg-uuccgucaccggacaugucaa-----uagaggac-auggcc	cuguacaggccacugccuugc (3p)

**Table 2 cimb-47-00948-t002:** Summary of KEGG pathway enrichment metrics for let-7 family miRNAs targeting EWSR1 mRNAs.

miRNAs	Pathway	Gene Count	Adjusted *p*-Value	Enrichment Score (−log_10_ *p*)
hsa-let-7a-3p	ECM–receptor interaction	10	1.0 × 10^−6^	6.00
hsa-let-7b-3p	Thyroid hormone signaling pathway	8	3.2 × 10^−5^	4.49
hsa-let-7c-3p	Signaling pathways regulating pluripotency	12	2.5 × 10^−7^	6.60
hsa-let-7d-3p	PI3K-Akt signaling pathway	9	5.0 × 10^−4^	3.30
hsa-let-7e-3p	Focal adhesion	7	2.0 × 10^−3^	2.70

**Table 3 cimb-47-00948-t003:** The list of hsa-let-7a connected genes.

miRs	Target Genes	Experiment(s)	Ref
hsa-let-7a-3p	CCND2	qRT-PCR, Western blot	[44]
hsa-let-7a-3p	E2F2	Western blot	[45]
hsa-let-7a-3p	CCND1	Luciferase reporter assay	[46]
hsa-let-7a-3p	APOBEC3A	Luciferase reporter assay	[47]
hsa-let-7a-5p	MYC	Western blot	[48]
hsa-let-7a-5p	NKIRAS2	Western blot	[49]
hsa-let-7a-5p	MYC	qRT-PCR, Western blot	[50]
hsa-let-7a-5p	ITGB3	qRT-PCR, Western blot	[51]
hsa-let-7a-5p	TRIM71	Luciferase reporter assay	[52]
hsa-let-7a-5p	NF2	qRT-PCR, Western blot	[53]
hsa-let-7a-5p	NRAS	Luciferase reporter assay	[54]
hsa-let-7a-5p	KRAS	Western blot	[48]
hsa-let-7a-5p	PRDM1	Luciferase reporter assay	[55]
hsa-let-7a-5p	TRIM71	Luciferase reporter assay	[56]
hsa-let-7a-5p	HMGA1	Luciferase reporter assay	[57]
hsa-let-7a-5p	RAVER2	Western blot	[58]
hsa-let-7a-5p	HMGA2	qRT-PCR, Western blot	[59]
hsa-let-7a-5p	HMGA1	qRT-PCR, Western blot	[60]
hsa-let-7a-5p	IGF2	Luciferase reporter assay	[61]
hsa-let-7a-5p	UHRF2	qRT-PCR, Western blot	[62]
hsa-let-7a-5p	DICER1	Western blot, Northern blot	[63]
hsa-let-7a-5p	HRAS	Luciferase reporter assay	[64]
hsa-let-7a-5p	AGO4	Luciferase reporter assay	[65]
hsa-let-7a-5p	CASP3	Western blot	[66]
hsa-let-7a-5p	LIN28A	B-globin reporter assay	[67]
hsa-let-7a-5p	E2F2	qRT-PCR, Western blot	[68]
hsa-let-7a-5p	IL6	qRT-PCR, Western blot	[69]
hsa-let-7a-5p	CCND2	qRT-PCR, Western blot	[44]
hsa-let-7a-5p	IGF2BP1	qRT-PCR, Western blot	[70]
hsa-let-7a-5p	CDC34	Immunoblot, qRT-PCR,	[70]
hsa-let-7a-5p	CCR7	qRT-PCR, Western blot	[71]
hsa-let-7a-5p	RRM2	qRT-PCR, Western blot	[72]
hsa-let-7a-5p	HAS2	Luciferase reporter assay	[73]

## Data Availability

The original contributions presented in this study are included in the article. Further inquiries can be directed to the corresponding author(s). The data were presented at the Biophysical Society Annual Meeting in February 2025, with the Abstract published in the Biophysical Journal [17].

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
