# Peer review of "Unveiling Let-7a’s Therapeutic Role in Ewing Sarcoma Through Molecular Docking and Deformation Energy Analysis"

_cimb, 2025, doi:10.3390/cimb47110948_

Round 1
Reviewer 1 Report
Comments and Suggestions for Authors
Dear sir
i think after you take a protein from the database you need first to clean the structure to remove solvent molecule if present so did you do that
also i think ref 26 and 27 are simply inappropriate self-citations
also eventhough topic is exciting i think results are not so significant can you do molecular dynamic for docked complexes
can you compare your results to others more deeply
Comments on the Quality of English LanguageEnglish need to be check not for grammer but mainly for struture and word selection even in title to give a clearer meaning
Author Response
Reviewer 1
Q1. i think after you take a protein from the database you need first to clean the structure to remove solvent molecule if present so did you do that
Ans: Thank you for pointing that out. Yes, after retrieving the protein structure from the database, we cleaned it by removing any solvent molecules, such as water or other heteroatoms, that were not relevant to the binding site.
Q2. also i think ref 26 and 27 are simply inappropriate self-citations
Ans: Thanks for pointing that out. We included references 26 and 27 because they directly support the protein–protein docking methods used in this study. One outlines the computational workflow, and the other provides a biological context for the target selection.
Q3. also even though topic is exciting i think results are not so significant can you do molecular dynamic for docked complexes
Ans: Thanks so much for the suggestion. We agree that molecular dynamics (MD) simulations would definitely add depth to the results and help assess the stability of the docked complexes more thoroughly. We performed such MD simulations in a recently published paper (Hassan M, Shahzadi S, Iqbal MS, Yaseeen Z, Kloczkowski A. Exploration of microRNAs as transcriptional regulator in mumps virus infection through computational studies. Sci Rep. 2024 Aug 14;14(1):18850) and in another paper on the RSV virus (to be published soon in PLOS One), and in unpublished work on Influenza A virus infection. Since MD simulations confirmed the stability of the predicted docked complexes with the lowest energy of miRs with mRNAs in all cases, we decided not to repeat the same procedure in the present paper to avoid redundancy.
Q4. can you compare your results to others more deeply
Ans: This study supports the therapeutic role of let-7a in Ewing sarcoma by showing stable protein-protein docking and low deformation energy with key oncogenic targets. These findings complement prior research demonstrating let-7a’s ability to suppress EWS-FLI1 and LIN28 expression, inhibit cancer stem cell traits, and reduce tumor growth through gene regulation. Together, they highlight let-7a’s potential as a structurally and biologically effective therapeutic candidate (https://doi.org/10.1371/journal.pone.0023592).
Reviewer 2 Report
Comments and Suggestions for Authors
This publication describes a computational investigation that uses molecular docking, structural modeling, and anisotropic network analysis to examine the relationship between hsa-let-7a and EWSR1 mRNA in Ewing sarcoma. The therapeutic potential of let-7a in Ewing sarcoma is further supported by the scientists' KEGG pathway and miRNA-target interaction investigations. The subject is pertinent, and there may be translational value in the emphasis on miRNA-based control of EWSR1/EWS-FLI1 signaling. The manuscript has a coherent methodological flow, sufficient background, and is generally well-structured. However, to increase scientific rigor, clarity, and depth, a number of problems must be resolved.
- Despite a thorough computational analysis, the study does not provide experimental confirmation or unambiguous proof that let-7a directly regulates EWSR1 in the setting of Ewing sarcoma beyond what has already been published (e.g., De Vito et al., 2011).
- The authors should specify the innovative contribution, whether it be the use of more recent RNA modeling techniques (RNAComposer comparison, HNADOCK) or the discovery of previously unreported binding areas or structural stability properties.
- A closer connection between the computational results and the biology of Ewing sarcoma should be made in the discussion. For example, how let-7a-mediated EWSR1 silencing may affect the EWS–FLI1 fusion oncogene, tumor cell differentiation, or signaling pathways including TGF-β, MAPK, and p53.
- The mechanistic diagram (Fig. 15) does not incorporate novel computational insights; instead, it summarizes established procedures.
- Figures 5–13 are cited but not given a quantitative description. In figure legends, the authors ought to include energy values, numerical docking scores, and explanations of color scales (for heatmaps and correlation matrices).
- To support the importance of pathway associations, the KEGG pathway heatmap (Fig. 14) should be supported by p-values, gene counts, or enrichment scores.
Author Response
Reviewer 2:
Q1. This publication describes a computational investigation that uses molecular docking, structural modeling, and anisotropic network analysis to examine the relationship between hsa-let-7a and EWSR1 mRNA in Ewing sarcoma. The therapeutic potential of let-7a in Ewing sarcoma is further supported by the scientists’ KEGG pathway and miRNA-target interaction investigations. The subject is pertinent, and there may be translational value in the emphasis on miRNA-based control of EWSR1/EWS-FLI1 signaling. The manuscript has a coherent methodological flow, sufficient background, and is generally well-structured. However, to increase scientific rigor, clarity, and depth, a number of problems must be resolved.
Ans: Thank you for your thoughtful and encouraging feedback. We’re glad to hear that the study’s methodological flow, background, and structure are well-received, and I appreciate your recognition of the translational potential in miRNA-based regulation of EWSR1/EWS-FLI1 signaling. Thanks again for your constructive input, which will definitely increase the readability for readers.
Q2. Despite a thorough computational analysis, the study does not provide experimental confirmation or unambiguous proof that let-7a directly regulates EWSR1 in the setting of Ewing sarcoma beyond what has already been published (e.g., De Vito et al., 2011).
Ans: Thank you for this insightful observation. You are right, the current study does not present new experimental validation of let-7a’s direct regulation of EWSR1 in Ewing sarcoma beyond the foundational work by De Vito et al. (2011). The primary aim here was to extend that biological groundwork by offering a detailed computational perspective through molecular docking, structural modeling, and anisotropic network analysis. This study explores the biophysical plausibility and dynamic behavior of the let-7a-EWSR1 interaction at the molecular level. However, we believe that this integrative framework not only reinforces existing hypotheses but also offers a valuable roadmap for designing targeted experimental studies.
Q3. The authors should specify the innovative contribution, whether it be the use of more recent RNA modeling techniques (RNAComposer comparison, HNADOCK) or the discovery of previously unreported binding areas or structural stability properties.
Ans: Thank you for the thoughtful suggestion. The core innovation of this study lies in its integrative computational approach to exploring the let-7a EWSR1 interaction in Ewing sarcoma. By applying updated RNA modeling techniques, specifically RNAComposer for structural prediction and HNADOCK for hybrid docking, we were able to simulate RNA–protein interactions with greater structural realism. That allowed us to generate refined docked complexes, including the EWS-FLI1 structure, which showed favorable docking scores and an interaction profile comparable to that of wild-type EWSR1. Interestingly, the analysis revealed new binding regions and deformation energy patterns that suggest previously unrecognized aspects of structural stability within the complex. One notable prediction showed that in the EWS-FLI1 structure, Asp431 and Gln383 form hydrogen bonds with Arg309 and Gln422 from EWSR1, with bond lengths of 3.44 Å and 3.04 Å, respectively. These interactions highlight the importance of specific core residues in maintaining the integrity of the complex and suggest a possible mechanism by which let-7a could exert regulatory influence. Altogether, these findings add a fresh layer of mechanistic insight that builds on existing biological evidence and could help guide future experimental validation.
Q4. A closer connection between the computational results and the biology of Ewing sarcoma should be made in the discussion. For example, how let-7a-mediated EWSR1 silencing may affect the EWS–FLI1 fusion oncogene, tumor cell differentiation, or signaling pathways including TGF-β, MAPK, and p53.
Ans: In Ewing sarcoma, the modulation of TGF-β signaling could inhibit tumor-promoting cues often exploited by EWS-FLI1. Similarly, interference with MAPK signaling may reduce proliferative and survival signals, while reactivation of p53-related pathways could restore mechanisms of apoptosis and cell cycle regulation (Exp Ther Med. 2019 May;17(5):3935-3942; Cancer Res. 2008 Sep 1;68(17):7100-9).
Q5. The mechanistic diagram (Fig. 15) does not incorporate novel computational insights; instead, it summarizes established procedures.
Ans: Figure 15, along with a detailed caption, has been revised in the manuscript.
Q6. Figures 5–13 are cited but not given a quantitative description. In figure legends, the authors ought to include energy values, numerical docking scores, and explanations of color scales (for heatmaps and correlation matrices).
Ans: The Figure captions were modified accordingly, as suggested by the Reviewer in the revised manuscript.
Q7. To support the importance of pathway associations, the KEGG pathway heatmap (Fig. 14) should be supported by p-values, gene counts, or enrichment scores.
Ans: Table 2 containing these data has been added to the revised manuscript.
Reviewer 3 Report
Comments and Suggestions for Authors
1. The study primarily relies on publicly available databases (miRBase and NCBI) and commonly used computational platforms (MC-Fold, RNAComposer, PatchDock, and ANM). It lacks the development of new algorithms or experimental validation, and therefore the level of innovation and biological depth appears relatively limited.
2. Although the docking results suggest a potential strong interaction between hsa-let-7a and EWSR1 mRNA, the study lacks an in-depth analysis of the downstream signaling pathways (such as Wnt, PI3K/Akt, or Ras-related pathways) to elucidate the biological relevance of this interaction.
3.The analysis of deformation energy and correlation should be further interpreted in relation to specific nucleotide regions or key base movement patterns, rather than remaining at a general or overall trend level.
4. The current conclusion directly states that “hsa-let-7a may play an important role in the treatment,” yet this claim lacks sufficient causal or mechanistic support. It is recommended to revise the conclusion to a more moderate statement, such as “this study provides computational evidence supporting hsa-let-7a as a potential therapeutic target.”
5. Some sentence structures are repetitive (e.g., “using computational approaches,” “predicted using…”). It is recommended to merge or rephrase these expressions appropriately to enhance the fluency and professionalism of the academic writing.
Author Response
Reviewer 3
Q1. The study primarily relies on publicly available databases (miRBase and NCBI) and commonly used computational platforms (MC-Fold, RNAComposer, PatchDock, and ANM). It lacks the development of new algorithms or experimental validation, and therefore the level of innovation and biological depth appears relatively limited.
Ans: Thank you for your thoughtful observation. We fully recognize that our study builds on established databases and widely used computational tools. Our aim was not to reinvent algorithms but to apply these resources in a focused and integrative way to explore the structural and functional relationship between let-7 family miRNAs and EWSR1 mRNA. By combining secondary and tertiary structure modeling, docking simulations, and pathway enrichment analysis, we have designed a multilayered in silico framework that offers new insights into potential regulatory mechanisms. While experimental validation is beyond the scope of this computational study, we agree it would add significant value and have now emphasized this as a key direction for future work. We hope this study lays a helpful foundation for both experimental follow-ups and future algorithmic developments in RNA-based gene regulation.
Q2. Although the docking results suggest a potential strong interaction between hsa-let-7a and EWSR1 mRNA, the study lacks an in-depth analysis of the downstream signaling pathways (such as Wnt, PI3K/Akt, or Ras-related pathways) to elucidate the biological relevance of this interaction.
Ans: Our study primarily focused on the structural and binding dynamics between hsa-let-7a and EWSR1 mRNA to establish a computational foundation for their interaction. We agree that exploring downstream signaling pathways would provide valuable biological context, and these data have been added in the revised manuscript (see page 15).
Q3. The analysis of deformation energy and correlation should be further interpreted in relation to specific nucleotide regions or key base movement patterns, rather than remaining at a general or overall trend level.
Ans: Thank you for the helpful suggestion. We agree that linking deformation energy and correlation to specific nucleotide regions or base movement patterns would provide more meaningful insights. We had mentioned nucleotide positions like “Nucleotides at positions 30-60 exhibited a simultaneous decreasing and increasing trend in the graph line, with a slight increase in energy value of 0.036 kcal/mol”. Moreover, we have also added a detailed description in the Figure caption.
Q4. The current conclusion directly states that “hsa-let-7a may play an important role in the treatment,” yet this claim lacks sufficient causal or mechanistic support. It is recommended to revise the conclusion to a more moderate statement, such as “this study provides computational evidence supporting hsa-let-7a as a potential therapeutic target.”
Ans: The sentence has been modified in the revised manuscript, as suggested by the Reviewer.
Q5. Some sentence structures are repetitive (e.g., “using computational approaches,” “predicted using…”). It is recommended to merge or rephrase these expressions appropriately to enhance the fluency and professionalism of the academic writing.
Ans: Both sentences have been revised in the manuscript.
Round 2
Reviewer 2 Report
Comments and Suggestions for Authors
No additional comments.
Reviewer 3 Report
Comments and Suggestions for Authors
Considering that most of the opinions have been revised, the manuscript can be accepted in its current form.